# Uncovering *in vivo* biochemical patterns from time-series metabolic dynamics

Yue Wu[1]☯, Michael T. Judge[2]☯, Arthur S. Edison[1,2,3,4]*, Jonathan Arnold[1,2,5,6]*

1 Institute of Bioinformatics, University of Georgia, Athens, GA, United States of America, 2 Department of Genetics, University of Georgia, Athens, GA, United States of America, 3 Complex Carbohydrate Research Center, University of Georgia, Athens, GA, United States of America, 4 Department of Biochemistry and Molecular Biology, University of Georgia, Athens, GA, United States of America, 5 Department of Statistics, University of Georgia, Athens, GA, United States of America, 6 Department of Physics and Astronomy, University of Georgia, Athens, GA, United States of America

☯ These authors contributed equally to this work.
* aedison@uga.edu (ASE); arnold@uga.edu (JA)

**Data Availability Statement:** Codes are available through GitHub (https://github.com/artedison/Edison_Lab_Shared_Metabolomics_UGA/tree/master/metabolomics_toolbox/code/net_ana). The experimental data can be found in Metabolomics Workbench (https://www.

## Abstract

System biology relies on holistic biomolecule measurements, and untangling biochemical networks requires time-series metabolomics profiling. With current metabolomic approaches, time-series measurements can be taken for hundreds of metabolic features, which decode underlying metabolic regulation. Such a metabolomic dataset is untargeted with most features unannotated and inaccessible to statistical analysis and computational modeling. The high dimensionality of the metabolic space also causes mechanistic modeling to be rather cumbersome computationally. We implemented a faster exploratory workflow to visualize and extract chemical and biochemical dependencies. Time-series metabolic features (about 300 for each dataset) were extracted by Ridge Tracking-based Extract (RTExtract) on measurements from continuous *in vivo* monitoring of metabolism by NMR (CIVM-NMR) in *Neurospora crassa* under different conditions. The metabolic profiles were then smoothed and projected into lower dimensions, enabling a comparison of metabolic trends in the cultures. Next, we expanded incomplete metabolite annotation using a correlation network. Lastly, we uncovered meaningful metabolic clusters by estimating dependencies between smoothed metabolic profiles. We thus sidestepped the processes of time-consuming mechanistic modeling, difficult global optimization, and labor-intensive annotation. Multiple clusters guided insights into central energy metabolism and membrane synthesis. Dense connections with glucose 1-phosphate indicated its central position in metabolism in *N. crassa*. Our approach was benchmarked on simulated random network dynamics and provides a novel exploratory approach to analyzing high-dimensional metabolic dynamics.

## 1 Introduction

Living organisms rely on a complex metabolic network, composed of thousands of metabolites and reactions [1]. Recent developments in experimental approaches [2–4] and feature

metabolomicsworkbench.org PR000738, DOI 10.
21228/M88X0P). Other related data can be found
in Supplementary materials.

**Funding:** National Science Foundation MCB-
2041546 (JA) National Science Foundation MCB-
1713746 (JA) National Science Foundation ERC-
1648035 (ASE) https://www.nsf.gov The funders
had no role in study design, data collection and
analysis, decision to publish, or preparation of the
manuscript.

**Competing interests:** The authors have declared
that no competing interests exist.

extraction [5] have enabled direct observation of complex metabolic dynamics, the ultimate
phenotypic response linking genes to metabolism [6, 7]. Here, we describe novel computa-
tional tools to extract biological knowledge from such high-dimensional metabolic time-series
datasets in a data-driven approach [8].

Time-series metabolomic datasets are rich and complex, but current statistical methods are
inadequate. Metabolic measurements often have hundreds to thousands of features, making it
a high-dimensional problem. Sampling as a function of time by novel tools, such as continuous
*in vivo* monitoring of metabolism by NMR (CIVM-NMR), further complicates the problem by
introducing dynamics for each feature [2]. Many features also have variable patterns in peak
locations and overlap. Fortunately, the latent dimensionality of the metabolome is constrained,
as changes in metabolites tend to be smooth over time, and dependencies exist between meta-
bolic features. Novel statistic methods utilizing the time-series dependencies are needed and
they will inform chemical identification and biological discoveries, which are important goals
of metabolomics.

One conventional approach is to model the metabolome explicitly. A metabolic network is
a dynamic system with metabolites (nodes) dependent on each other through reactions and
regulation (edges). Direct modeling of such a network often involves flux balance approaches
(FBA) [9] or, when dynamics are needed, simulation of ordinary differential equations
(ODEs) [10–12]. Such direct approaches face challenges in computation and data accessibility.
Including a realistic topology and making appropriate parameter estimations are computa-
tionally expensive [10, 11, 13–15]. Compromises in topological structure (e.g., regulation) can
result from a lack of pathway knowledge or enough data and significantly reduce a model's
utility. Specifically, many compounds cannot be observed or confidently annotated and quan-
tified, and useful ODE solutions might not be discovered due to the additional uncertainty [2,
4, 16]. Typical time-series metabolomic measurements leave most nodes in metabolic net-
works unobserved because of limitations in detection and annotation [2].

Our alternative approach enables interactive exploration of metabolic dynamics and
extracts biological and chemical information from the time series. Based on the framework of
functional data analysis (FDA), we smoothed the time-series metabolic features and projected
them into lower dimensions to visualize the dominant metabolic trends [17–19]. We also
enabled a quick comparison of experiments under different perturbations and revealed meta-
bolic adaptations. Furthermore, we identified networks and clusters based on correlation and
dynamical associations between time series separately [20–24]. The correlation network
expanded chemical annotation and the empirical dynamic network revealed *in vivo* biochemi-
cal functions [1]. Our workflow thus prioritizes NMR features for further biological
investigation.

## 2 Results

### 2.1 Workflow to analyze time-series NMR spectra

In this study, we built a workflow to enable knowledge discovery from the new type of data,
time-series NMR. CIVM-NMR measures *in vivo* metabolism through time, producing rich
and complex spectral data. Such datasets can reveal the underlying dynamic metabolic process,
profile metabolic adaptations to perturbations, facilitate annotations and uncover biochemical
regulation. These are accomplished through dimensionality reduction and network
construction.

The data were collected continuously on the model filamentous fungus, *N. crassa*, which
lived in the NMR probe for about 12 hours (Fig 1A) [2]. We recorded NMR spectra of the liv-
ing organism as it metabolized, which led to a complex spectral surface with coordinates in

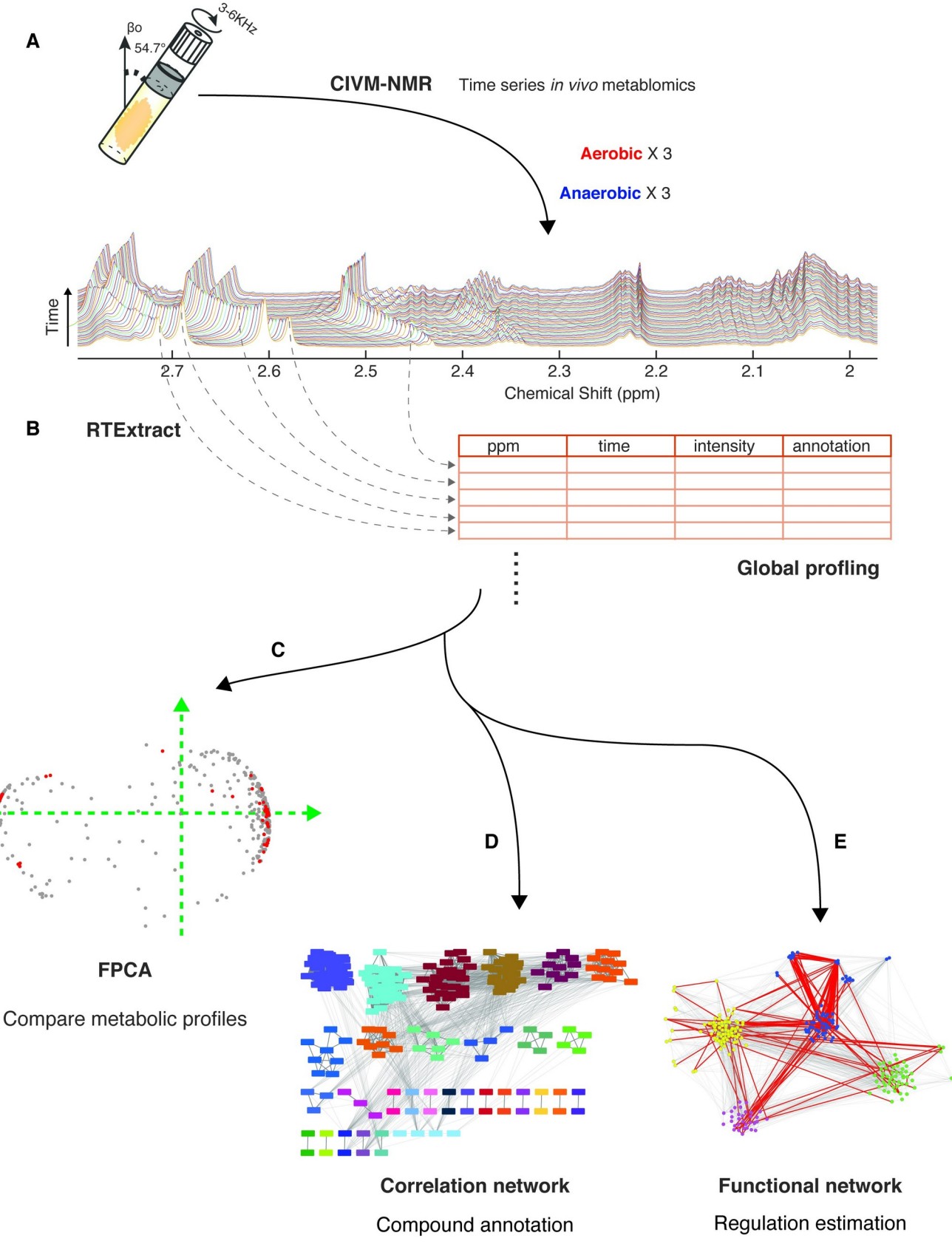

**Fig 1. The analytic workflow for time-series NMR data.** A: time-series *in vivo* measurements were collected through CIVM-NMR under different conditions, including oxygen accessibility (aerobic vs anaerobic) and different carbon sources. B: RTExtract efficiently extracted peak information and enabled a global untargeted metabolic profiling (~300 time-series features/sample). The data were analyzed by multiple approaches (C-E). C: Dimensionality reduction using FPCA compared metabolic trends under different conditions. D: A correlation network provided annotations. 34 clusters were found by correlating the time series. E: Functional groups and regulation were estimated through dependencies between time-series features. Four clusters were found through CausalKinetiX and community clustering of the *in vivo* time series. Detailed information can be found for each approach: C (Figs 2 and 3 and S2 and S3), D (Figs 4 and S4) and E (Figs 5 and S5).

parts per million (ppm) for the NMR axis and hours for the time axis (Fig 2A). In this study, we worked with experiments of different carbon sources and labeling: six experiments feeding on glucose (three of an aerobic condition and three of an anaerobic condition) [2] and two experiments feeding on $^{13}$C uniformly labeled pyruvate in an aerobic condition (S2 and S3 Files).

Each time-series dataset was processed in a global and untargeted way through RTExtract (Fig 1B) [5]. About 300 metabolic features were extracted for each dataset, consisting of around $10^5$ data points per experiment. RTExtract efficiently quantified time-series NMR spectra even for peaks with overlap and pH-induced chemical shift changes [2, 5].

We first projected the metabolic features into a lower dimension to visualize differences in dynamics (Fig 1C). Compounds were grouped, showing different biochemical processes (Figs 2 and S2). The same compound was also compared under different conditions to show responses to perturbation (Figs 3 and S3). We then constructed networks based on time series and clustered them to search for chemical and biological associations in the living sample (Fig 1D and 1E). The correlation network produced 34 clusters and seven validated compound annotations without requiring extraction or 2D experiments (Figs 1D and 4 and S4). This can also include those compounds that were consumed and below detection level at the extraction time point. The functional network presented *in vivo* metabolic processes, including central energy metabolism and phospholipid metabolism (Figs 1E, 5 and 6 and S5).

## 2.2 Dimensionality reduction of metabolic dynamics

Time-series NMR spectra have high dimensionality, and the patterns are diverse for different metabolites (Fig 2B). Projection into lower dimensions through the functional principal component analysis (FPCA) visualizes different groups and trends between metabolites and between conditions.

We first compared metabolic features in aerobic (Fig 2B) and anaerobic conditions (S2A Fig). In both conditions, the first two principal components (PCs) explained the major variance (PC1 about 80%) (Figs 2B and S1 and S2B). The PC1 eigenfunction indicates an increasing trend added upon the mean curve of all metabolic features (black curves in Figs 2B and S2C; Details in Methods). Metabolites that were consumed appear as negative values along PC1, while those that were produced have positive values along the same axis. PC1 captures the dominant metabolic trend of consuming carbohydrates (e.g., glucose and trehalose) and producing amino acids (e.g., tyrosine and alanine) and fermentation products (e.g., ethanol). The PC2 eigenfunction is orthogonal to PC1 and has a decreasing and then increasing pattern (Fig 2B), which captures more complex dynamics including glucose 1-phosphate (G1P) and choline. PC2 accounts for more variance in the aerobic condition (13%) than in the anaerobic condition (8%) (Figs 2B and S2C), in agreement with the inspection of the NMR spectra [2]. Most metabolites have multiple NMR peaks, and these are clustered together in the score plot, demonstrating the stability of FPCA separation (Figs 2B and S2A). For example, glucose nodes are clustered and similar to the trends of trehalose (Fig 2B).

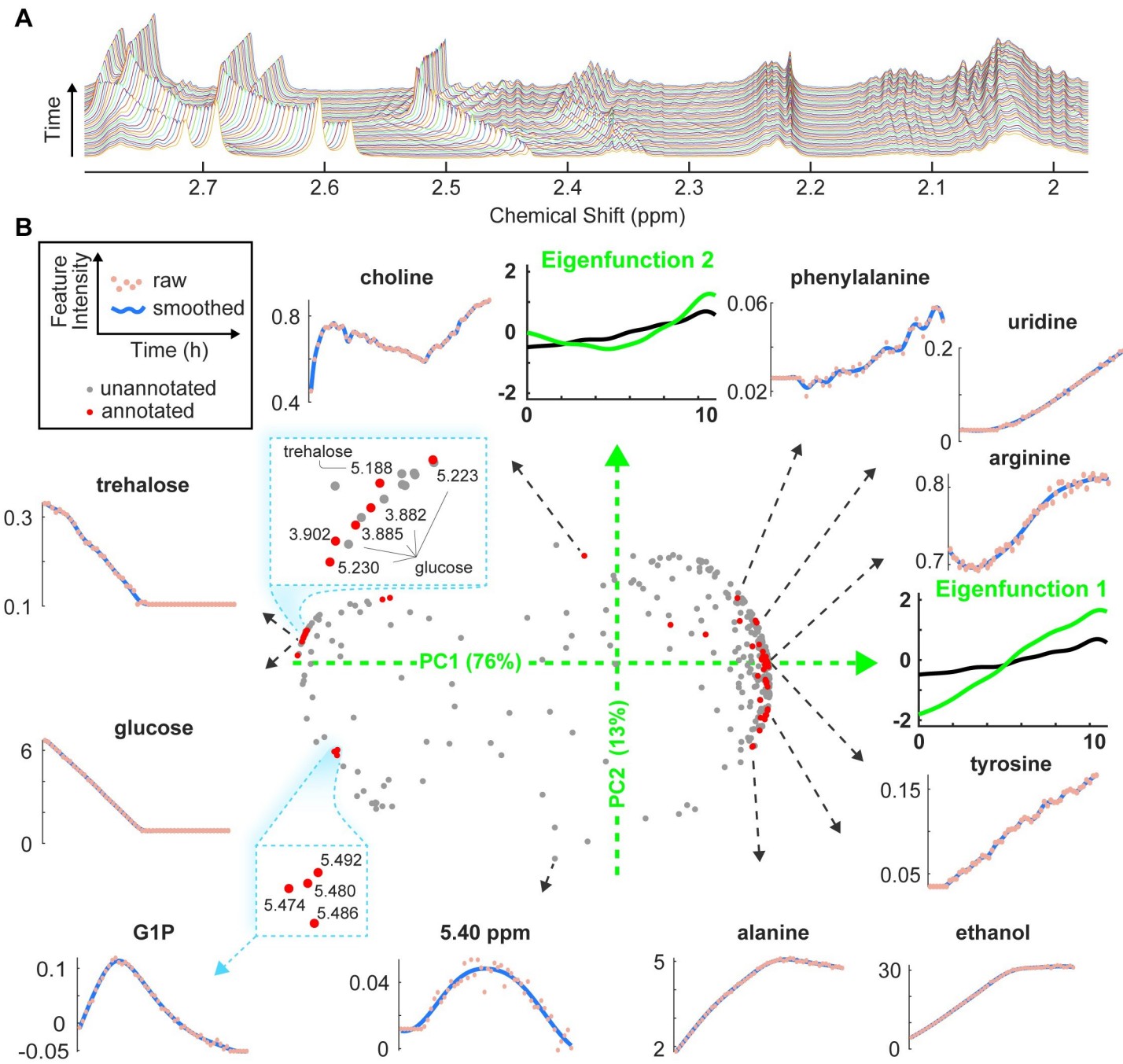

**Fig 2. Dimensionality reduction through FPCA captures dominant variations in metabolic dynamics within individual samples.** A: An example CIVM-NMR spectral region of *N. crassa* under the aerobic condition. The stack plot shows changes in a region of the NMR spectra through time. Each time point is distinguished by a different line color. B: Time series of *N. crassa* NMR features collected under aerobic conditions were smoothed and visualized in two dimensions. In the middle panel, each NMR feature trajectory (unannotated, grey; annotated, red) is summarized as a combination of the two dominant eigenfunctions (green axes) which define the axes of the FPCA scores plot. The X (Y) axis represents scores for PC 1 (2), and the variance percentage of each PC is given in parentheses. Eigenfunctions in FPCA are analogous to loading vectors in multivariate PCA. While loading vectors are often presented as effects of each feature, eigenfunctions are presented as the smoothed effect added to the non-constant mean curve. The green curves represent adding a fraction (square root of eigenvalue) of the corresponding eigenfunction to the mean curve of all NMR features (black curve). Selected time-series features are presented around the scores plot (raw NMR peak intensities, points; FDA-smoothed intensity profiles, blue lines). In the scores plot, two inset figures (blue boxes) show details of glucose and G1P clusters with chemical shifts of the NMR features. All nodes in the G1P cluster belong to G1P. Each NMR feature was mean-centered and scaled by standard deviation before the FPCA analysis. Time is in hours. Percentages of explained variance of different PCs are presented in S1 Fig. Results for one anaerobic dataset can be found in S2 Fig. Details on preprocessing, smoothing and FPCA can be found in Methods.

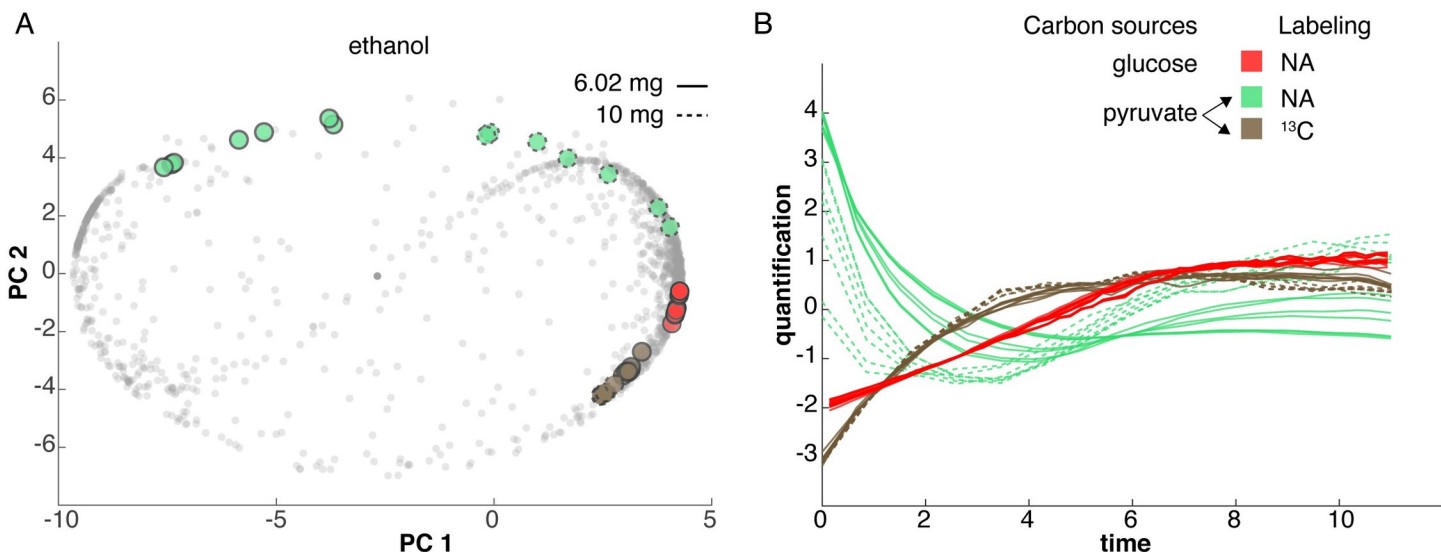

**Fig 3. Comparing ethanol profiles under different growth conditions through FPCA.** Two different carbon sources were compared: natural abundance glucose (3 experiments) and uniformly $^{13}$C-labeled pyruvate (2 experiments). The glucose experiments were all done at a high density (10 mg/63 μL), and the pyruvate experiments were done at low (6mg /63 μL, solid lines) and high (10 mg/63 μL, dashed lines) densities. The ethanol features from the glucose experiments are shown in red. The $^{13}$C-labeled ethanol produced in the $^{13}$C-pyruvate experiments is shown in brown. The unlabeled ethanol produced in the $^{13}$C-pyruvate experiments is shown in green. A: FPCA score plot indicates the overall patterns of ethanol in each of these experiments. Each point represents one ridge in one sample. The small, grey points correspond to all the other ridges detected in these experiments. The X (Y) axis represents scores for PC 1 (2). B: Time trajectory (hours) of the highlighted features from A. Each curve was centered and scaled for A and B. Similar plots for other compounds can be found in S3 Fig.

Time series in the four FPCA quadrants represent different trends in metabolism (Fig 2B). Curves with positive PC1 and negative PC2 increased and plateaued, and they correspond to compounds related to fermentation and storage (e.g., alanine and ethanol). Curves with positive PC1 and PC2 continued to increase over time (e.g., phenylalanine and uridine), and several amino acids are in this group. Curves with negative PC1 and positive PC2 decreased and plateaued (e.g., glucose and trehalose). These carbohydrates were consumed until they fell below our detection limits of about 50 μM in CIVM-NMR $^1$H spectra [2]. Other curves with relatively high magnitude in PC2 values had more nonlinear patterns (e.g., choline and the unknown feature at 5.4 ppm), indicating more complex metabolic processes. In addition to the general patterns represented by each quadrant, the scores themselves also followed a continuous trend as the variance percentages of the PCs change between points (Fig 2B). For example, the arc of scores from alanine to phenylalanine mirrored the changes in corresponding curves as PC2 gradually increased from negative to positive.

## 2.3 Compare metabolic experiments through FPCA

The FPCA projection can be expanded to the comparison of metabolic profiles under different experimental conditions. We can visualize metabolic adaptations by comparing the dynamics of the same compound with different media or mutants. To illustrate the flexibility of this approach, we compared five different CIVM-NMR datasets, all grown aerobically (Figs 3 and S3). Three used natural abundance glucose as the carbon source and two used uniformly $^{13}$C-labeled pyruvate as the carbon source. The glucose replicates were all grown at a high density of about 10 mg per rotor volume [2]. The pyruvate replicates were grown at two different densities: one at 10 and the other at 6 mg per rotor volume.

In an isotopic labeling study, CIVM-NMR allows to simultaneously monitor both labeled and unlabeled metabolites in one experiment [2]. We accomplished this by interleaving a

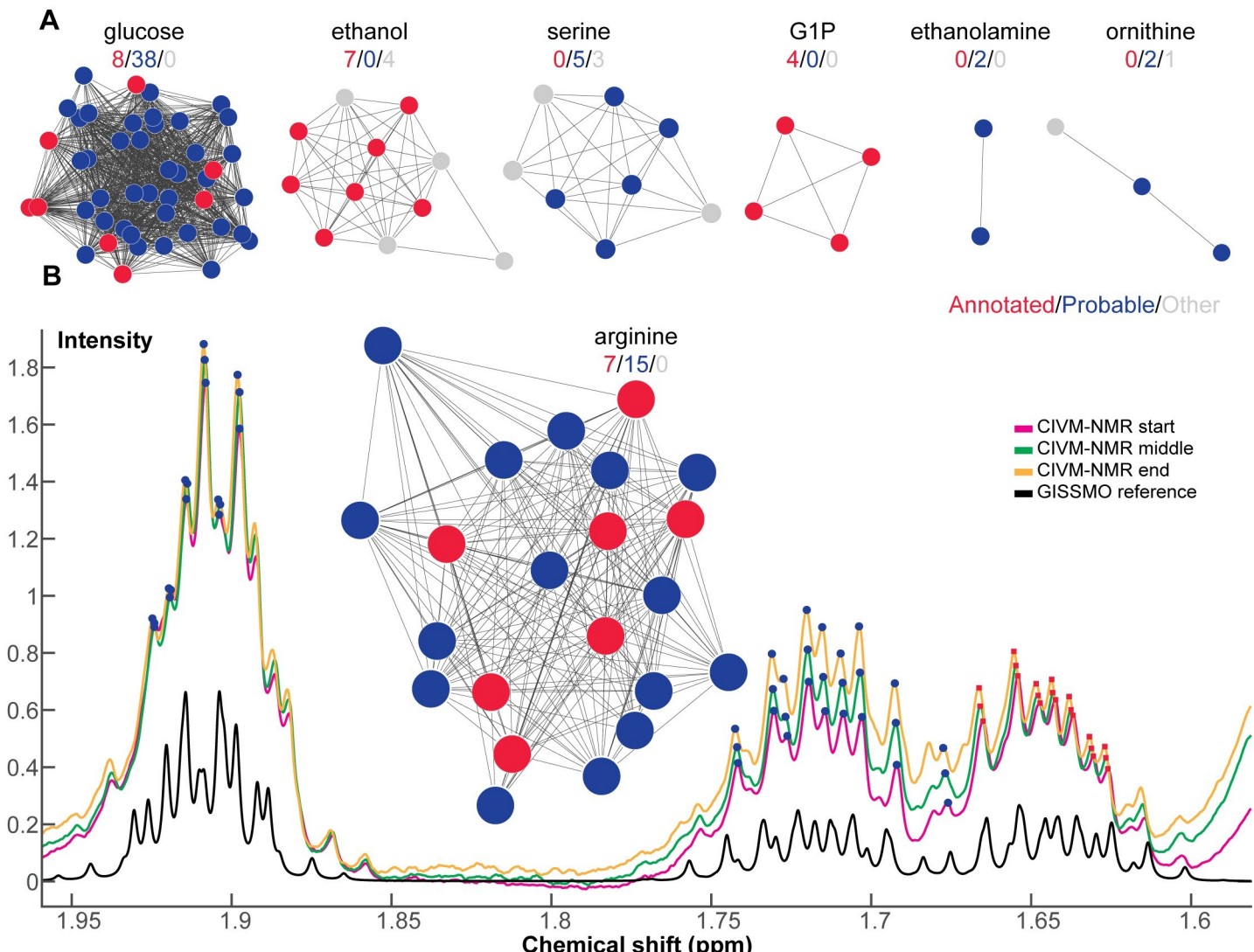

**Fig 4. Clusters in correlation networks contribute to annotation.** A correlation network was built using Spearman correlation between time-series features in the glucose feeding experiments, and clusters were discovered (More details in Methods). Clusters in the correlation network agreed with previous annotations. A: Selected clusters are co-visualized with annotations from our prior publication [2]. Some compounds can be directly annotated by correlation network clustering. Red nodes represent NMR features with prior annotations [2]; blue nodes represent NMR features that were consistent with reference peaks from the assigned compound but could not be confidently annotated due to spectral overlap; gray nodes represent other unannotated NMR features. The numbers above each cluster indicate the number of nodes for each type. B: The cluster for arginine is co-visualized with experimental and reference spectra. The X (Y) axis represents chemical shift (intensity). The pink-, green-, and yellow-colored spectra are from three representative time points in the CIVM-NMR dataset. The black spectrum is the same region of the GISSMO [26] reference spectrum for arginine. Red squares (blue circles) represent features with confident (probable) annotation to arginine and correspond to the color in the network cluster. The entire clustered CN is displayed in S4 Fig.

standard 1D $^1$H experiment that detects all $^1$H atoms in the sample and a 1D $^{13}$C-HSQC that selects only $^1$H atoms with directly bonded $^{13}$C (Details in Methods). This approach enables us to detect different pools of the same metabolite that originate from different metabolic pathways, as the organism is unlabeled at the start of the isotopic labeling experiment.

We compared patterns of the same compounds under different conditions in FPCA and showed corresponding extracted ridges for the same compound (Figs 3 and S3). The pyruvate consumption pattern is similar to that of glucose (S3 Fig). The ethanol features from high-density cultures all fell on the positive PC 1 axis, regardless of carbon source and labeling. In

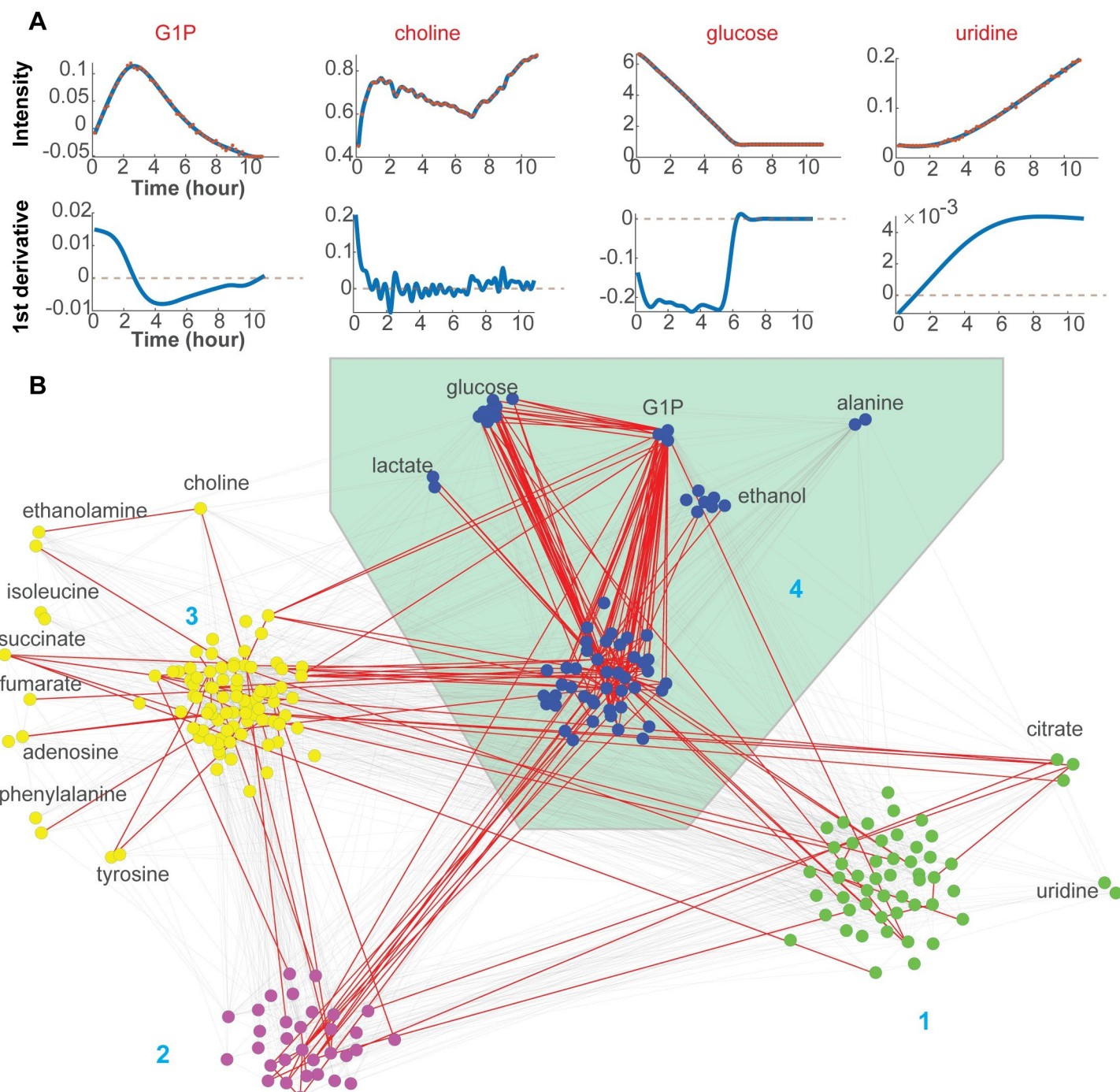

**Fig 5. Analyzing metabolic dynamics from derivatives in the glucose feeding experiments.** A: A few examples corresponding to points in Fig 2B are visualized. Intensities (row one) and corresponding 1st derivatives (row two) are presented. The X (Y) axis represents time (value). The blue curve represents a smoothed curve from FDA, and the red points represent raw measurements. Dotted grey lines indicate zero derivatives. B: An empirical metabolic network and clusters were inferred from time series. Time-series features were summarized into four clusters through CausalKinetiX and community clustering (More details in Methods). Each node represents one time-series feature, and each edge represents one inferred link from CausalKinetiX. Red edges are supported through bootstrapping at the cutoff of 40%. Clusters are distinguished by color. Cluster 4 is overrepresented by metabolites in central energy metabolism and highlighted with a green background. Some nodes are attached with annotations and were manually moved outside for better visualization. Bootstrapping details can be found in S5 Fig and Methods.

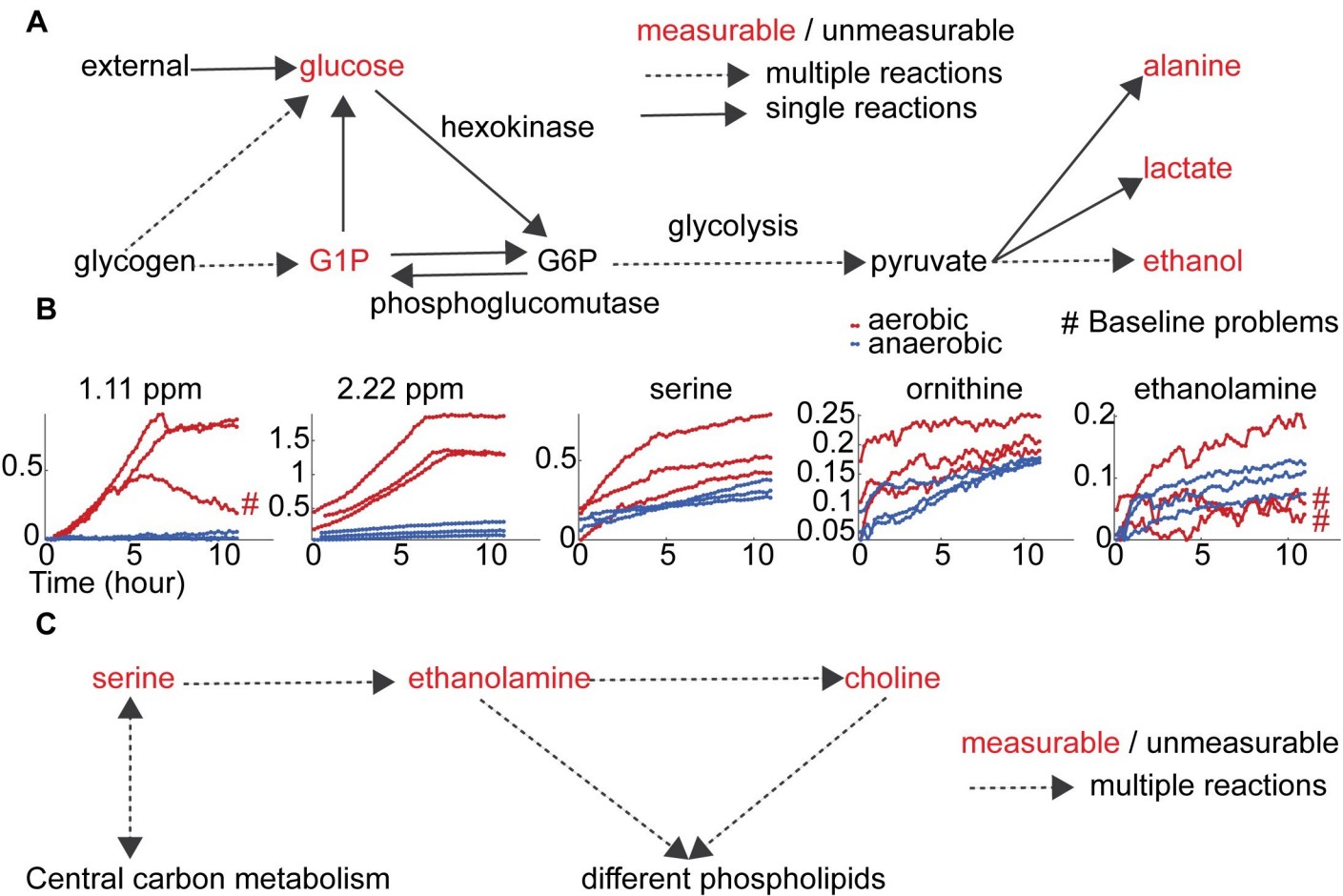

**Fig 6. Biochemical connections for central energy metabolism and membrane synthesis in the glucose feeding experiments.** A: Hypothesized pathway through the central energy metabolism: glycolysis, glycogen degradation and fermentation. Measured (unmeasured) compounds are visualized in red (black). Steps with multiple (single) reactions are represented by dotted (solid) lines. B: Time-series quantification for selected features: high degree unannotated features in Fig 5B and compounds related to phospholipid synthesis and amino acid metabolism. The X (Y) axis represents time (means of scaled ridge intensities). Red (blue) represents aerobic (anaerobic) conditions. The '#' symbol indicates that the quantified peaks are highly affected by regional baseline changes. C: The hypothesized pathway for phospholipids synthesis.

contrast, the unlabeled ethanol from the low-density pyruvate culture is negative on the PC 1 axis, clearly showing a density dependence of ethanol production (Fig 3). The ethanol produced in the high-density glucose cultures is tightly clustered for all the replicates.

The ethanol in the $^{13}$C-labeled pyruvate cultures has interesting dynamics. In both densities, the unlabeled ethanol (green labels, Fig 3) is on the positive PC 2 axis, and it was first consumed and then produced after 3 or 6 hours for the high and low densities, respectively (Fig 3B). In contrast, the $^{13}$C-labeled ethanol patterns (brown labels, Fig 3) are identical for the two culture densities: they increased and then plateaued at about 4 hours. A similar pattern exists in glucose culture, where ethanol plateaued when the major carbon source, glucose, was exhausted.

## 2.4 Expand metabolite annotation through a correlation network

Among approximately 300 features in each CIVM-NMR dataset, about 60 features (20 compounds) were annotated at the expense of time-consuming experiments and expert labor in

our original study [2]. We note that due to the nature of the CIVM-NMR experiment and RTExtract algorithm [5], our extracted features are individual components of J-coupled multiplets, which should have perfect linear correlations in ideal cases. With the extracted ridges, additional information can be directly obtained through clustering a correlation network (CN) of ridge intensities without any 2D experiments. Similar to statistical correlation spectroscopy (STOCSY) [24, 25], we constructed a CN using the Spearman correlation of extracted CIVM-NMR ridges in the 6 glucose feeding experiments, found 30 well-separated clusters in the CN, searched each cluster in 1D database and validated candidates for annotation (Figs 4 and S4; Details in Method).

Metabolite identification and database matching are always challenging in metabolomics [27, 28]. Our lab [29] and others [30] proposed different confidence scales for annotation. The traditional annotation approach [29] was based on metabolomic sample extraction, 2D NMR experiments (heteronuclear single quantum coherence spectroscopy, HSQC; Total Correlation Spectroscopy, TOCSY) and COLMARm [31], which was used in the original CIVM-NMR publication [2]. Such a process is standard and powerful but has drawbacks when the goal is to annotate and quantify ridges in time-series NMR. First, besides the additional work of extraction and 2D NMR, the sampling time point(s) for extraction depends on the dynamics of the feature of interest, and multiple time points might be needed. The compound of interest needs to exist for COLMARm annotation, and metabolites that decrease over time in a CIVM-NMR run may not be detectable in a sample extracted at the end of the run. Second, sample extraction will differentially enhance or diminish some compounds and introduce chemical shift changes depending on the extraction solvent. Changes in relative peak intensity and chemical shift complicate the next step, and mapping from COLMARm to CIVM-NMR spectra is not perfect. CN clustering of CIVM-NMR ridges covers compounds through the whole time range, needs no further experiments or mapping, and has no extraction biases. CN clusters provide a practical way to augment annotation and can be used with 2D NMR at specific time points. We also defined a new confidence scale for annotations for CIVM-NMR datasets (Details in Method) [2, 29].

Among the 30 clusters, we presented seven examples here (Figs 4 and S4). Glucose, ethanol, G1P and arginine were annotated through both mapping from COLMARm (Fig 3 in CIVM-NMR publication) [2] and CN clusters. Glucose, ethanol, and arginine were at level 4 (See Methods), and G1P was raised to level 5 with spike-in validation (S9 Fig). CN clusters provided more peaks for quantification (blue nodes in Fig 4A). Those peaks were not assigned in our previous publication because of overlaps with other compounds in the extracted sample. However, in CIVM-NMR data, those peaks were highly correlated and clustered with high confident peaks (red nodes). A detailed example of such an assignment is presented for arginine with experimental NMR spectra and the database standard (Fig 4B). 2D NMR of the extracted sample indicates lysine and leaves arginine only quantifiable for a subset of peaks at around 1.65 ppm (red points in Fig 4B) [2]. However, the CN cluster of other peaks (blue points) with confidently annotated peaks (red points) indicates the low concentration of lysine that allows for reliable quantification of arginine (Figs 4B and S4).

In addition to expanding the peak assignment for annotated compounds, three CN clusters with blue points but no red points (Figs 4A and S4) revealed new compound annotations (e.g., serine, ethanolamine and ornithine) [2]. These compounds appeared in 2D experiments on methanol-extracted samples but could not be mapped and quantified in the CIVM-NMR study because of overlap, noise, and poor peak shapes [2]. Particularly, relative compound concentration and spectral overlap pattern in an extracted sample were different from those in CIVM-NMR spectra, and mapping the two is not trivial. Nonetheless, CN revealed highly correlated clusters of CIVM-NMR peaks for those compounds (level 4).

## 2.5 Derivatives of metabolic profiles provide additional insights into metabolic dynamics

Besides chemical associations, biological associations can also be uncovered from the high-dimensional dataset. The first derivative of the ridge intensities with respect to time yields rates of change in metabolite concentrations and provides further information on metabolic states (Fig 5A). Rates (Fig 5A, the second row) can be visualized and co-analyzed with corresponding intensity curves (Fig 5A, the first row). Derivative-based analysis in the following two sections will focus on the glucose feeding experiments. Glucose was consumed at a relatively constant rate and fell below the limit of detection at around six hours [2]. Uridine accumulated throughout the experiment but had two distinct intervals between which the rate changed at about six hours when glucose was exhausted. In the first six hours, the rate of uridine accumulation increased; after six hours, that rate was constant. The intensity and first derivative of G1P were complex: it was initially produced (positive rate); the intensity reached a maximum just after two hours when it started to decrease; consumption reached a maximum rate at around four hours. Choline accumulated rapidly in the first hour, and its rate remained relatively low afterward. However, the first derivative estimation of choline is relatively noisy, and a higher smoothness penalty (See Methods) might improve the estimation after two hours. We expanded the analysis of associations between ridge intensities and rates through model searching and network construction in the next section.

## 2.6 Define metabolic associations through network clusters

In theory, one could individually analyze all 300 NMR curves and search for biological relationships. Not only would that be inefficient, but it would miss critical patterns that are inherent to the interconnected metabolic pathways, in which the rate of change of one compound often depends on concentrations of several other compounds. Through integrating different perturbations and network construction, these *in vivo* metabolic connections and regulation can be discovered. We used CausalKinetiX and community clustering [22, 23] to search for the model. We built a stable and predictive network from CIVM-NMR datasets collected under aerobic and anaerobic conditions and with glucose as the carbon source (Fig 5B). We assumed that the network topology is stable and that different conditions will lead to changes in the edge strengths through reaction or regulation [23, 32].

The estimated functional network (FN) edges indicate associations between rates and concentrations, which can be biochemical reactions or regulation. The performance of edge estimation depends on the number of perturbation conditions (S1 File and S6–S8 Figs), and the estimation can be noisy with fewer conditions. Hence, we relied on the clustering of undirected networks to improve signal detection and to find clusters representing different biochemical processes. Four clusters were recovered through community clustering, among which Cluster 4 is well-supported by bootstrapping (Figs 5B and S5; More details in Methods) and includes several critical metabolites in central energy metabolism in *N. crassa* (Fig 6A). Glucose, G1P and ethanol were consistently grouped into one cluster in bootstrapping (S5 Fig). Metabolic features with inverse trends, primary carbon source (e.g., glucose) and product (e.g., ethanol, lactate and alanine), were clustered together (Figs 2B and 5B and S5), delineating the primary flux under the experimental conditions. Cluster 3 is relevant to amino acid and phospholipid metabolism, and more time-series data with perturbations, such as mutations, are needed for cleaner separation. In the FN, most peaks annotated to the same compound are consistently in the same cluster (Fig 5B). Confidence in our estimations of network edges and clusters was assessed using a benchmark dataset (S1 File and S6–S8 Figs).

## 3 Discussion

Our exploratory workflow to uncover metabolic associations consists of dimensionality reduction and network analysis by both correlation and CausalKinetiX. FPCA enables metabolic profile comparison, the CN expands incomplete annotation, and the FN discovers metabolic connections, which is important in most practical cases, where pathway knowledge is incomplete. Our workflow prioritizes metabolic clusters for further isotopic labeling and perturbation experiments, as part of the discovery cycle of system biology [8].

### 3.1 Uncover *in vivo* biochemical functions in central energy metabolism

FN Cluster 4 was found to be associated with central energy metabolism, containing glucose, lactate, G1P, ethanol and alanine (Figs 5B and S5). In the glucose feeding experiments, glucose was consumed, and fermentation products accumulated. This is largely represented by PC1 in FPCA (Figs 2B and S2) and agrees with our previous publication [2]. Additionally, several biochemical connections were found as follows.

G1P can indicate the functional level of glycolysis and central energy metabolism. We saw dense connections between G1P and most other NMR features in the reconstructed FN (Fig 5B and S1 Table). G1P can be produced by glycogen degradation and converted to glucose or glucose 6-phosphate (G6P) (Fig 6A). G6P is an intermediate metabolite in glycolysis, which is central in many metabolic branches. Except for glucose, glycolysis metabolites (e.g., G6P and pyruvate) were below our detection limit, preventing direct observation of glycolysis. The reaction between G1P and G6P is catalyzed by phosphoglucomutase and is reversible (Fig 6A) [2, 33, 34], so G1P can be an alternative indicator of glycolysis fluxes, explaining its high associations with many NMR features in the FN. Additionally, the two transition points (indicated by changes in rate) of G1P at about two and four hours did not coincide with glucose exhaustion at six hours (Fig 5A), indicating alternative regulation mechanisms. Specifically, the initial accumulation of G1P can be caused by rate limitation in glycolysis during a transition from starving to high rates of glucose consumption [2, 34]. Based on the FN analysis, our working hypothesis is that the metabolic associations with G1P depend on phosphoglucomutase (gene id: *NCU10058*) and its reversible connection to glycolysis. In the future, we will test the effects of its mutants (strain id: FGSC #18976) on central energy metabolism.

We also found that alanine might serve as an alternative carbon source after glucose exhaustion. Alanine is in the increase-and-plateau quadrant in the FDA scores plot but has a slight decreasing pattern after glucose exhaustion at about six hours (Fig 2B). In FN, it is in Cluster 4 and has dense links with glucose (Fig 5B and S1 Table). Alanine is directly connected to pyruvate through a transamination reaction and can be used for carbon and nitrogen storage when they are abundant [2, 35]. The observation that alanine consumption started when glucose was exhausted, suggests the hypothesis that alanine was an alternative carbon source in our experiments.

Diverse patterns of ethanol under various conditions (Fig 3) were uncovered through metabolic profile comparison by FPCA. We were able to compare different carbon sources (glucose and pyruvate), different starting densities (6 or 10 mg per 63 μL) and isotopic labeling. In the $^{13}C$-pyruvate data, we were able to monitor two distinct pools of ethanol, one that was isotopically labeled with $^{13}C$ and the other that was at natural abundance carbon. These have very different patterns, and the distribution on the FPCA scores plot allows us to understand their relationship. For example, the unlabeled ethanol data from the $^{13}C$-pyruvate experiments was initially consumed and then released after 2–6 hours, depending on density. But the $^{13}C$-labeled ethanol in the same $^{13}C$-pyruvate experiments increased and then plateaued at about 4

hours. Similar comparisons could be made with studies of other genotypes, carbon sources, or isotopic labeling.

Interesting unannotated features were selected for further experiments. Many high-degree nodes in FN Cluster 4 (Fig 5B) were not annotated [2]. Many of them belong to glucose or are considerably affected by glucose concentration because of overlap (Fig 4A). Among other unannotated nodes, features at 2.22 ppm and 1.11 ppm have consistent FN connections and different dynamics between aerobic and anaerobic conditions (Figs 5B and 6B and S1 Table). Both nodes are functionally connected to glucose and G1P, and 2.22 ppm is also connected to fermentation products, including ethanol and lactate. These connections are supported by bootstrapping. Both features increased and plateaued in aerobic conditions and have a much lower level in anaerobic conditions. The peak at 2.22 ppm also exists in both unlabeled and $^{13}$C-labeled forms in the pyruvate dataset. We suspect that the 2.22 ppm feature corresponds to an N-acetyl functional group and are working on its full annotation.

## 3.2 Fluxes towards membrane synthesis and amino acid metabolism

Phospholipid synthesis was also active in our experiments. Three compounds related to this process were annotated: serine, ethanolamine and choline (Figs 2B, 4A and 6B). The former two were newly annotated here by searching CN clusters using the COLMAR 1D databases [36]. Serine is associated with glucose in the FN (Fig 5B and S1 Table). Its synthesis starts from 3-phospho-D-glycerate, a glycolysis intermediate, and its degradation produces pyruvate. Serine can flow to phospholipid synthesis, ethanolamine (and phosphatidylethanolamine) and then to choline (and phosphatidylcholine) (Fig 6C) [33, 37, 38]. Genes *chol-5*, *chol-8*, *chol-11* and *chol-12* were previously found related to the first step, and *chol-1* was for the second step [39]. An association between ethanolamine and choline was also found in the FN. Both choline and ethanolamine are important precursors for phospholipid and membrane synthesis. Different dynamics of the two precursors might indicate different fluxes to corresponding membrane phospholipids. Changes in phospholipid ratio during development have been observed previously [40], and the formation of vacuoles in stressed (e.g., starving) *N. crassa* cells is well-known [41].

Ornithine was also newly annotated and quantified in CIVM-NMR spectra through searching CN clusters in the COLMAR database [36]. It was connected to valine and tyrosine in the FN, though its quantification is relatively noisy. Connection to glutamate was expected [38] but not found, and this might result from glutamate's intense connections to many other amino acids and dependencies on them. The FN connection to valine and tyrosine might indicate balance and regulation among different amino acids.

## 3.3 Technical improvements on network-based dynamic analysis

Even though our prior analysis did not require extensive pathway mapping, including known pathway knowledge can yield further biological insights. Specifically, the empirically estimated FN clusters can be compared or merged with conventional metabolic pathways for interpretation. Starting from compounds of interest, possible paths can also be searched in conventional pathways and compared with those in the empirical FN [42, 43]. New developments in graph neural networks can also be applied to merge information from the FN networks and pathways [44].

Including more perturbation experiments can considerably improve the FN method (S1 File). Edges, directions and clusters can be more confidently estimated when more diverse perturbations are available. Although CIVM-NMR and RTExtract simplify measuring time-series metabolic features [2, 5], collecting many perturbations is still expensive and time-consuming. Metabolic dynamics highly depend on both the media condition and prior culturing process

[2], so a refined experimental procedure is needed to maintain consistency between perturbation experiments. Effective exploration of interesting perturbations is also crucial, and the results in this paper provide valuable next targets.

CN clustering provides a simple initial annotation of the CIVM-NMR dataset, complementing the traditional approach involving extraction and 2D experiments. Currently, we have searched CN clusters in the COLMAR database to expand annotation [36]. However, many features remained unannotated because of the considerable overlap in NMR spectra and limited coverage in the database. We can improve this by combining CN connections with a probabilistic graph model. Biochemical associations in FNs and metabolic pathways also help reduce the searching space of possible compounds.

Integration of our different analyses (FPCA, CN and FN) and perspectives can facilitate the utility of our tools by researchers with broad backgrounds. We are currently working on an efficient web interface to integrate these analyses [45]. In the new interface, with a few clicks, biologists will be able to select one cluster in the FN or the CN and highlight nodes in the other network, visualize network clusters in the FPCA scores plot or vice versa, and search for interactions between a group of network nodes in metabolic pathways [42, 43, 46]. Based on biochemical pathways, information from other omics experiments, such as transcriptomics [7] and proteomics [8] can also be co-analyzed with metabolomics [15].

We created a framework to summarize the dominant trends of both annotated and unannotated features and enabled comparisons between perturbation experiments. We then leveraged the unique properties of these data to facilitate and expand annotation. Finally, we integrated this feature set into a network of functional dependencies stable across conditions to yield biological insights. This work helps bridge the gap between untargeted time-series metabolomic data and biochemical integration.

## 4 Methods

### 4.1 *Neurospora* culturing, preparation, and data collection

Briefly, Neurospora bd 1858 mycelia were grown for around 30 hours in 3% Glucose Vogel's minimal media in 50 mL shaking flask cultures under constant light [2]. An hour before the start of the CIVM-NMR experiment, mycelia were poured with media into a 50 mL conical tube and transported to the NMR facility. Working quickly, a small piece of mycelium was pulled from the mycelial mass and washed four times in 1.5 mL conical tubes containing 1 mL minimal media without carbon sources. The mycelium was pat-dried, weighed and adjusted to around 11 mg, then resuspended in 500 μL NMR media (1.5% glucose). The total volume of mycelia and media was adjusted to around 63 μL, and contents were transferred to a 63 μL (4 mm) zirconia HR-MAS rotor, which was capped with a breathable cap. All experiments were collected on a 600 MHz Bruker NEO equipped with a 4-mm CMP-MAS probe running Top-Spin (v4.0.1; Bruker). Details in sample growth, preparation, data collection, and preprocessing are described in [2]. For 13C-pyruvate experiments, the same procedure was carried out with the following adjustments:

1. No citrate was used in the Vogel's media.

2. Mycelia were grown in 1.5% Sucrose instead of 3% Glucose.

3. The total mass of the rotor contents was carefully adjusted to 50 mg ≈ 50 μL (mycelia and media) by removing/adding media.

4. Immediately before recording, pyruvate addition was carried out by swapping 10 μL rotor liquid for 10 μL concentrated 13C-pyruvate (220 mM; Aldrich 490717-500MG, Lot #

FMBBC3492) in the NMR media for a final concentration of 37 mM in the rotor. Thus, most pyruvate metabolism was observed as early as possible. An unavoidable delay due to rotor spin-up (~3–10 min) still occurred.

5. The spinning speed was reduced from 6000 Hz in the glucose experiments to 3500 Hz for pyruvate samples [2].

6. Data were recorded and averaged every eight scans (~35 s) for noesypr1d and every 32 scans (~30 s) for hsqcetgsisp2.2 experiments. These were collected in an interleaved manner as described previously and smoothed by moving average [2].

## 4.2 NMR feature extraction

We followed the original method in RTExtract [5] to extract NMR time-series features. The experimental data were collected with 52 time points, under aerobic and anaerobic conditions, and each condition contains three experiments. Comprehensive regions of interest (ROI) were selected and tracked in each dataset. Some NMR features existed and were extracted in only part of the time range. The empty values in chemical shift and intensity were filled with those of neighboring existing time points. This is used instead of intensity zero filling because there is often a local baseline shift in the spectra.

NMR features were matched across different experiments. Annotated features were matched directly, and unannotated features were matched based on chemical shift differences. In the latter circumstance, distances were calculated between features in different samples and the closest pairs (maximum threshold 0.01 ppm) were matched. Each NMR feature in one sample is represented by its mean chemical shift through time.

For $^{13}$C-labeling experiments, we interleaved standard $^{1}$H 1D (noesypr1d) with $^{13}$C-HSQC 1D (hsqcetgpsisp2.2). The $^{1}$H 1D data provides information on all metabolites, and the $^{13}$C-labeled species have addition peaks from the large $^{1}$H to $^{13}$C coupling constants. The $^{13}$C-HSQC 1D data selects for only $^{1}$H atoms that are directly bound to $^{13}$C, which significantly simplifies the data but also eliminates the species that are not isotopically labeled. Our RTExtract algorithm works equally well for either the $^{1}$H 1D or $^{13}$C-HSQC 1D datasets. Direct comparison of them needs normalization based on the large $^{13}$C-labeled pyruvate signals. Natural abundance pyruvate only has a single peak from the methyl group at about 2.37 ppm in the $^{1}$H 1D spectrum. This single peak is split into 4 peaks in $^{13}$C-pyruvate because of the large 1 bond and smaller 2 or 3 bond $^{13}$C-$^{1}$H J couplings. We used $^{13}$C decoupling during the $^{13}$C-HSQC 1D acquisition so that the dataset only has a single peak at about 2.37 ppm. To calculate the normalization factor, we set the intensity of the single peak in the $^{13}$C-HSQC 1D spectrum equal to the sum of the intensities of the 4 pyruvate peaks in the $^{1}$H 1D spectrum at each time point. The normalization factor was calculated for pyruvate peaks higher than 10% quantile and then averaged. This same normalization factor was then applied to all other compounds in the datasets. Afterward, all ridges were scaled by the maximum of the unlabeled compound and then scaled within the experiment and labeling. This last step reduced variance from multiplets and simplified visualization.

## 4.3 Smoothing and dimensionality reduction

Time-series features were then analyzed with FDA, which is a collection of methods for analyzing curves or functions, including time series. The analysis often starts from smoothing and then can go into multiple directions, including derivative analysis, dimensionality reduction and regression [17–19]. FPCA and derivative-based regression [23] were used in our workflow.

Time-series curves were first fitted with B-splines with smoothness penalties (Eq 1 as the objective function) [17, 18]. The first part of $F$ is the sum of squared distance between the original data value $y_j$ and the smoothed function $x(t_j)$. The second part is the smoothness penalty, where $D$ is the derivative operator. The 1$^{st}$ derivatives represent rates of metabolic activities and were forced to be smoothed, making the smoothness penalty based on the 3$^{rd}$ derivatives (Eq 1) [18]. The penalty parameter $\lambda$ was searched in a log scale on 33 values from $10^{-4}$ to $10^4$ and chosen by minimizing generalized cross validation [47].

$$F = \sum_j [y_j - x(t_j)]^2 + \lambda \int [D^3 x(t)]^2 dt \tag{1}$$

Dimensionality reduction was done with FPCA [17, 18] based on smoothed curves. Just as PCA provides lower-dimensional representations of the original dataset, FPCA provides a similar smoothed representation of time series. Like a PC vector in PCA, each found FPCA eigenfunction (harmonic, $\xi(t)$) is orthonormal ($\int \xi_i(t)\,\xi_j(t)dt = 0$ where $i \neq j$ and $\int \xi_i^2(t)dt = 1$) and explained the maximal variances iteratively. Each time-series feature is mean-centered, scaled by standard deviation and smoothed before FPCA. The origin is represented by the mean function (curve) with the mean spline coefficients from all curves. Algorithm details of FPCA can be found here [17, 18].

## 4.4 Correlation network estimation and visualization

A correlation network was constructed from experimental NMR measurements to help compound annotation. Spearman correlations were calculated between time-series features concatenated from different experiments, and the largest 10% were included in the network. The network was clustered by the Markov Clustering Algorithm (MCL) based on correlation value and granularity parameter 5 [48]. Chemical shifts of each network cluster were searched through the GISSMO database [26] and COLMAR 1D Query [36] for possible matches. The GISSMO searching process is automatic, with Cytoscape controlled through R by RCy3 [49, 50], GISSMO search through API (http://gissmo.nmrfam.wisc.edu/peak_search) and spectral visualization in MATLAB.

## 4.5 New annotation confidence level for CIVM-NMR experiments

We defined new confidence levels to accommodate the CIVM-NMR experiment and CN clustering [2, 29]. This new definition includes the requirement of mapping back to CIVM-NMR spectra and the annotation power of CN clusters. The levels are defined from 1 to 5 with increasing confidence. (1) There is a similarity of 1D $^1$H spectra (between a standard reference and CIVM-NMR spectra at any timepoint). (2) There is HSQC match using COLMAR [51] from an extracted sample and the compound can be mapped back to CIVM-NMR spectra. (3) Compound annotation can be found for a CN cluster in CIVM-NMR spectra. (4) There is double matching from two sources as well as matching in CIVM-NMR spectra. It can be HSQC match and TOCSY/HSQC-TOCSY validation from extracted data using COLMARm [31]. It can also be HSQC match and CIVM-NMR CN cluster validation. (5) The compound is spiked into the CIVM-NMR sample and validated.

## 4.6 Functional network estimation and clustering

In a metabolic network, nodes depend on each other (e.g., reactions or regulation), and densely connected subnetworks represent specific functions. We estimated associations

(edges) between metabolic features (nodes) by CausalKinetiX [23] and searched for functional groups by community clustering [22]. CausalKinetiX can estimate predictive and stable edges in a metabolic network in a time-efficient manner [23]. Unlike conventional methods for extracting time-series dependencies [52], CausalKinetiX learns directly from non-stationary time series in which relationships between rates and concentrations hold up under different perturbations [32]. Features were extracted by RTExtract as in the previous section [5].

CausalKinetiX [23] estimates edges, the dependencies between the changing rate of one compound and the concentration of some compound(s) (Eq 2, adapted from Eq 1 in [23]). $X^i(t)$ is the time-series features in experiment $i$. $T$ is the target variable feature, and $S$ is a subset of features. The dependency $f$ holds under all experiments $i$ (Eq 2). Predicative and stable models [23] were searched for derivative estimation of each target feature ($X_T$). Covariate features ($X_S$) were ranked by importance, and connections with p-values less than 0.05 were selected as estimated edges. Before fitting, target variables were smoothed with a penalty on the 3$^{rd}$ derivatives [23]. To reduce computational complexity, the expected number of terms in each model was set to at most two, and pre-screening was used [23]. Interaction terms were also included. The model was fitted in the derivative mode. Other parameters were default values [23]. Constructed networks were undirected and clustered by community-based clustering based on topology [22] using clusterMaker in Cytoscape [53, 54].

$$\frac{dX_T^i(t)}{dt} = f\left(X_S^i(t)\right) \tag{2}$$

Clustering stability was evaluated through bootstrapping. The dataset was resampled 100 times, and, in each iteration, the complete time series for each feature were sampled with replacement among the three experiments of fixed conditions. The network was then constructed and clustered as above based on each new bootstrapped dataset. Frequencies that two features share the same cluster were calculated as bootstrapping support.

## 4.7 Benchmarking dataset simulation

We simulated random networks and corresponding dynamics as a benchmark. Networks and clusters were estimated based on the simulated time series and compared with the ground truth. We generated different random networks with 100 nodes and three clusters (sizes: 40, 20 and 20). Edges were randomly generated for node pairs with higher probability within clusters (0.15) and lower probability among all nodes (0.015).

Temporal dynamics were simulated by ODEs, which were generated based on the random networks with nodes (edges) representing metabolites (reactions or regulation). The reaction was formulated by a sum of regulated mass actions (Eq 3). $X(t)$ are time-series features, where $T$ indicates the target variable features, and $r$ indicates the regulatory features. Multiple reactants ($1 \leq N_h \leq 2$) are involved in one reaction, and each feature is connected by multiple reactions ($N$). Kinetic parameter $k$ is regulated by $X_r$ with the exponent $\alpha \in \{-1, 1\}$. $s$ is the stoichiometric factor. Reaction direction was randomly generated, including reversible reactions. Reactions with multiple reactants ($N_h > 1$) were simulated by randomly combining single reactions. The ODEs were simulated under different initial conditions ($X(0)$) as different experimental perturbations. The simulation time grids were set from 0 to 5 with the step size

0.2. Kinetic parameters and initial conditions were uniformly sampled ($U(0,1)$).

$$\frac{dX_T(t)}{dt} = \sum_j^N [s_j \, k_j^* \prod_h^{N_h} X_h(t)] \tag{3}$$

$$k^* = k \prod_r X_r^{\alpha_r}(t)$$

Some procedures were introduced to ensure similarities between simulation and experimental measurement. Duplicated signals belonging to the same compounds were simulated to resemble multiple peaks in NMR. For each times series in the ODE simulation, a random number of signals ($N_S \in \{1,2,3,4,5\}$) were added with each multiplication factor uniformly sampled ($U(0.3,3)$). For each fixed ODE set and initial condition, three replicates were simulated with Gaussian noise (Eq 4) [23]. *Y(t)* is the observation, *X(t)* is the simulation with no noise, and $\sigma(X)$ is the standard deviation function. The partial observation was also simulated to resemble the incomplete coverage of experimental measurements. 50% of the features were randomly selected to be observable.

$$Y(t) = X(t) + \sigma^* \tag{4}$$

$$\sigma^* \sim N(0, a)$$

$$a = 0.02 \cdot \sigma(X) + 10^{-7}$$

## 4.8 Performance evaluation on the benchmark dataset

Edge estimation was evaluated by recall and precision (Eqs 5 and 6). $N_{tp}$ is the number of true positives, $N_{pp}$ is the number of predicted positives, and $N_{cp}$ is the number of observable conditional positives. For compound features with multiple signals, the instances were counted for each compound.

$$Precision = \frac{N_{tp}}{N_{pp}} \tag{5}$$

$$Recall = \frac{N_{tp}}{N_{cp}} \tag{6}$$

The capability of recovering underlying clusters was evaluated by the match ratio (Eq 7). Among all estimated clusters, the one with the most overlapped nodes with the real cluster was selected. $N_{overlap}$ is the number of overlapped nodes, and $N_{cluster}$ is the size of the estimated cluster. For compound features with multiple signals, the instances were counted in terms of each signal.

$$R = \frac{N_{overlap}}{N_{cluster}} \tag{7}$$

## 4.9 Code and availability

The program was implemented in MATLAB, R and Python. Codes are freely available through GitHub (https://github.com/artedison/Edison_Lab_Shared_Metabolomics_UGA/tree/master/metabolomics_toolbox/code/net_ana). The experimental data can be found in Metabolomics Workbench (https://www.metabolomicsworkbench.org PR000738). The local programs were

implemented under R 3.5.1, MATLAB_R2018b and Cytoscape 3.8.0 on macOS 10.15.7. The extensive simulation was implemented on HPC: R/3.5.0-foss-2019b on Sapelo2 at Georgia Advance Computing Resource Center (GACRC).

## Supporting information

**S1 Fig. Percentages of explained variance in FPCA in one aerobic dataset.** The X (Y) axis represents the number of PC (percentage of variance). The first two PCs, especially PC1, explain the most variance. The corresponding scores plot and eigenfunction can be found in Fig 2B.
(PDF)

**S2 Fig. Dimensionality reduction captures dominant variation in metabolic dynamics in one anaerobic dataset.** A: The first two PC dimensions are visualized, and dominant changing patterns are presented. Each point represents the time series of one NMR feature, and some of them are highlighted with compound annotations (red). The X (Y) axis represents scores for PC 1 (2). B: Percentages of explained variance are presented for the first few PCs. The X (Y) axis represents the number of PC (percentage of variance). The first two PCs, especially PC1, explain most variances. C: Eigenfunctions are plotted for PC1 and PC2. The middle black curve represents the mean time series; the red curve represents the effects of adding a fraction (square root of eigenvalue) of the corresponding eigenfunctions to the mean curve. The X (Y) axis represents time (value). The percentages of variance explained are presented in parentheses. NMR features were centered and scaled before the PCA analysis. Results for aerobic conditions can be found in Figs 2 and S1.
(PDF)

**S3 Fig. Additional comparison of metabolic profiles under different carbon sources through FPCA.** Two different carbon sources were compared: natural abundance glucose (3 experiments) and uniformly [13]C-labeled pyruvate (2 experiments). The glucose experiments were all done at a high density (10 mg/63 μL), and the pyruvate experiments were done at low (6mg /63 μL, solid lines) and high (10 mg/63 μL, dashed lines) densities. The chemical features from the glucose experiments are shown in red. The [13]C-labeled metabolites produced in the [13]C-pyruvate experiments are shown in brown. The unlabeled metabolites produced in the [13]C-pyruvate experiments are shown in green. A: FPCA score plot indicates the overall patterns of different compounds in each of these experiments. Each point represents one ridge in one sample. The small, grey points correspond to all the other ridges detected in these experiments. The X (Y) axis represents scores for PC 1 (2). The score plot was kept the same with different compounds highlighted. B: Time trajectory (hours) of the highlighted features from A. Each curve was centered and scaled for A and B. C: Detailed comparison of ethanol in the [13]C-pyruvate experiments. Two experiments with different amounts of organisms are visualized by different shapes. [13]C-labeled and unlabeled ethanol are distinguished by colors. Normalization was applied to make them comparable. Ridges of triplets around 1.1 ppm were used to quantify ethanol in C, and all tracked ridges were used in Fig 3A and 3B.
(PDF)

**S4 Fig. Clustering of the correlation network helps compound annotation.** A correlation network was built upon time-series features in the glucose feeding experiments, and clusters were found (More details in Methods). Each cluster is highlighted by a different color and assigned one number. Single nodes are in blue. Details of specific clusters are shown in Fig 4.
(PDF)

**S5 Fig. Consistency of functional networks was evaluated by clustering frequency.** Colors in the heatmap represent relative frequencies that two NMR features share the same clusters in bootstrapping. Each row or column represents one feature. Red (white) indicates more (less) co-occurrences of the two features. The top bar indicates different compounds by colors. The dashed box highlights nodes of glucose, ethanol and G1P that are frequently presented in the same cluster. Diagonal values are not presented, as the same feature will always be in the same cluster. The bootstrapping results are also visualized in the network in Fig 5B.
(PDF)

**S6 Fig. Time-series dynamics were simulated from random networks.** One example random network and corresponding dynamics are presented. A: The random network (with 100 nodes) was simulated with clusters. Red, green and blue nodes indicate three simulated clusters where internal links are denser than inter-cluster links. Gray nodes do not belong to any clusters. B: Time-series dynamics were simulated for each node under different initial conditions. The X (Y) axis indicates time (mean simulated value). Trajectories of three different nodes under three different conditions are presented. The mean value was calculated from three different replicates with the same nodes and conditions but different random noise. More detail on random networks and dynamics simulation can be found in Methods and S1 File.
(PDF)

**S7 Fig. Model performances were benchmarked on a simulated dataset with partial observation.** Our workflow was evaluated on a simulated benchmark dataset with partial observation. In the random network, each node (100 in total) represents one compound, and each cluster is a set of compounds with denser inner cluster connections. Time dynamics were simulated based on the networks through ODEs under different initial conditions. A subset of the time-series features was observable and clustered (More details in Methods). The performance in estimating edge and recovering clusters was evaluated. A: Recall and precision (Y-axis) for edge estimation are presented under different numbers of initial conditions (X-axis). Edge estimation improves with more conditions. B: The performance in cluster recovering is presented for the three clusters under a different number of conditions. The X (Y) axis represents the number of conditions (match ratio). The match ratio is the proportion of nodes from the matched real cluster in the best-recovered cluster (More details in Methods). The red dotted lines indicate the match ratio of a random group of nodes (baseline). The estimated clusters can recover more nodes from real clusters, and the performance improves with more conditions. The error bars represent two standard errors calculated from the simulation and reconstruction of 59 random networks. Simulated random networks and example time series can be found in S6 Fig. Simulation-based evaluation with redundant signals can be found in S8 Fig. The performance on an experimental dataset can be found in Figs 5B and S5.
(PDF)

**S8 Fig. Model performances were benchmarked on a simulated dataset with partial observation and redundant signals.** Our workflow was evaluated on a simulated benchmark dataset with partial observation and redundant signals. In the random network, each node (100 in total) represents one compound, and each cluster is a set of compounds with denser inner cluster connections. Time dynamics were simulated based on the networks through ODEs under different initial conditions. Time-series features were also expanded and scaled by random factors as an analogy of multiple peaks corresponding to the same compound in NMR. A subset of the time-series features was observable and clustered (More details in Methods). The performance in estimating edge and recovering clusters was evaluated. A: Recall and precision

(Y-axis) for edge estimation are presented under different numbers of initial conditions (X-axis). Edge estimation improves with more conditions. B: The performance in cluster recovering is presented for the three clusters under a different number of conditions. The X (Y) axis represents the number of conditions (match ratio). The match ratio is the proportion of nodes from the matched real cluster in the best-recovered cluster (More details in Methods). The red dotted lines indicate the match ratio of a random group of nodes (baseline). The estimated clusters can recover more nodes from real clusters, and the performance improves with more conditions. The error bars represent two standard errors calculated from the simulation and reconstruction of 59 random networks. Simulated random networks and example time series can be found in S6 Fig. Simulation-based evaluation with no redundant signals can be found in S7 Fig. The performance on the experimental dataset can be found in Figs 5B and S5.
(PDF)

**S9 Fig. The G1P Spiking experiment showed a level 5 annotation.** Multiple contrasting experiments were collected through time: minimal media [2], minimal media and 10 mg mycelia, spiking glucose in the culture, and spiking G1P in the culture. The X-axis represents chemical shift. The arrow shows the G1P peaks, which increased after G1P spiking.
(PDF)

**S1 Table. Neighbors of annotated peaks in central energy metabolism.** Neighbors of selected driver nodes in the CausalKinetiX network were listed (Fig 5B). The searching starts from glucose, G1P, ethanol, alanine, and lactate. Columns include driver (the searching start node), annotation (the annotation of the neighbor), ppm (chemical shifts of the neighbor nodes), edge_support (whether the edge is supported by bootstrapping), and cluster (which cluster the neighbor node belongs to).
(XLSX)

**S1 File. Performance evaluation based on benchmarking dataset.** Edge and cluster estimation was evaluated on a simulated benchmark dataset.
(DOCX)

**S2 File. Summary of ridge tracking result of pyruvate dataset.**
(MAT)

**S3 File. Summary of ridge tracking result of the original CIVM-NMR dataset.**
(MAT)

## Acknowledgments

We thank Abby Moore, Chen Hsieh and Robert Powers for useful discussions. We appreciate computational support from GACRC, especially Shan-Ho Tsai and Zhuofei Hou. Jeremy Zucker and Bill Cannon at Pacific Northwest National Laboratory provided a metabolic database, Hesam Dashti and Hamid Eghbalnia provided assistance with the GISSMO NMR library, and Niklas Pfister provided technical details on CausalKinetiX.

## Author Contributions

**Conceptualization:** Yue Wu, Michael T. Judge, Arthur S. Edison, Jonathan Arnold.

**Data curation:** Michael T. Judge.

**Formal analysis:** Yue Wu.

**Funding acquisition:** Arthur S. Edison, Jonathan Arnold.

**Methodology:** Michael T. Judge, Arthur S. Edison, Jonathan Arnold.

**Project administration:** Jonathan Arnold.

**Resources:** Arthur S. Edison, Jonathan Arnold.

**Software:** Yue Wu.

**Supervision:** Arthur S. Edison.

**Validation:** Arthur S. Edison.

**Visualization:** Yue Wu.

**Writing – original draft:** Yue Wu, Michael T. Judge.

**Writing – review & editing:** Arthur S. Edison, Jonathan Arnold.

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
