## [Decision Letter · Decision Letter 0]

2 Feb 2022

PONE-D-21-25919Uncovering in vivo biochemical networks from time-series metabolic dynamicsPLOS ONE

Dear Dr. Arnold,

Thank you for submitting your manuscript to PLOS ONE. After careful consideration, we feel that it has merit but does not fully meet PLOS ONE’s publication criteria as it currently stands. Therefore, we invite you to submit a revised version of the manuscript that addresses the points raised during the review process.

Specifically, both reviewers found the work potentially interesting but they raised a significant number of issues that must be addressed before we can further proceed.

We look forward to receiving your revised manuscript.

Kind regards,

Oscar Millet

Academic Editor

PLOS ONE

https://journals.plos.org/plosone/s/file?id=ba62/PLOSOne_formatting_sample_title_authors_affiliations.pdf"

Reviewers' comments:

Reviewer's Responses to Questions

**Comments to the Author**

1. Is the manuscript technically sound, and do the data support the conclusions?

Reviewer #1: Partly

Reviewer #2: Yes

2. Has the statistical analysis been performed appropriately and rigorously? 

Reviewer #1: I Don't Know

Reviewer #2: Yes

3. Have the authors made all data underlying the findings in their manuscript fully available?

Reviewer #1: Yes

Reviewer #2: Yes

4. Is the manuscript presented in an intelligible fashion and written in standard English?

Reviewer #1: No

Reviewer #2: No

5. Review Comments to the Author

Reviewer #1: This study involves the development of a new analytical workflow to visualise complex NMR-based metabolomic data from fungus cultures. The approach has the potential to reveal new metabolic signatures and speed processing so is widely but the manuscript is written in a technical manner lacking key details that would make it approachable to a more general reader. Better annotation of Figures (particularly the supplemental information) and restructuring of manuscript (clear separation of legends, results narrative and discussion commentary) would also improve readability and accentuate the impact for the general audience of this journal.

I appreciate that the focus here is on the data processing but the fungal cells have a relatively simple metabolome. To validate the downstream impact of such an approach it would be reassuring to see analysis of additional data from a complex cell line or human cell system where different cells within a mixture may exhibit metabolic variation. Alternatively data from Neurospora cultured with a metabolic stressor/altered culture medium conditions to check that data arising make biological sense

The analysis is based on a published data set generated using cultures under aerobic and anerobic conditions. Even though the data is preexisting it would be important to provide more information on the approach in the methods section (eg source of fungus and cultural conditions/densities, metabolite extraction procedure, annotation method) since fungal life stage will impact on outcome under different conditions. Similarly the advantage here is the ability of the analysis to track analytes over time and pick up peaks unassigned by other approaches. Thus it may be worth including a schematic diagram or summary of the analytical workflow to highlight the key advantages of the system.

Figure 1 – the stack plot here is a nice way to visualize the temporal change in spectra. Would it be possible to annotate some of the key peaks (glucose, ethanol, G1P etc) and show a similar composite image for spectra collected under anaerobic conditions? This would help with assessment of the latter Figures. In particular it would be helpful to see the arginine plot on this figure so that it can be compared with the output in Figure 2

Is there choline in the media used to grow the cells?

The ridges within the trace form the basis of the cluster analysis. Does this introduce a bias for detection of more abundant metabolites even though aim is to maximize visibility by scanning over a timeframe?

Could the approach be combined with isotopic metabolic tracer analysis using labelled precursors to confirm that the hypothesized fluxes (Fig 4) are detected in the time resolved analysis? Otherwise the discussion of this data is a little speculative.

Much useful information is supplied in supplemental figures but these are barely referred to or describes in places. File S1 is a word document describing benchmarking. Not sure how this works as a supplemental information source?? Should it be badged as supplemental methods??

Reviewer #2: This manuscript describes an approach to improve identification of metabolites in longitudinal metabolic profiles using correlation networking. The approach is sound and this is an area of intensive developments.

The document is written in a manner that is difficult to follow. Some sections are very detailed when not really necessary while the introduction is very sparse. The draft would greatly benefit from a Figure 1 with a proper explanation of the whole procedure that would guide the reader and also help the structure of the text. For instance when discussing the expansion of the annotation I found confusing the part where annotations are found because of correlation or because of look up into COLMAR. I understand that a correlation STOCSY or even covariance NMR approach produces a TOCSY-like spectra that can be compared with real spectra, but in the text it is not clear what type of data is input of what process. Is the CN used to build a spectra as an input into COLMAR or is the network directly compared with other networks obtained from spectra? As this is not clear, it makes it difficult to understand what data is annotated using COLMAR and what data is annotated by CN prior to COLMAR. Page 9, lines 188 to 196 are very difficult to follow.

The next step is also described in a confusing manner, is the functional network used to improve the annotation or is the previous clustering used to improve the outcome of the functional networking? I guess the former, but then the authors fails (in my opinion) to discuss how this form of clustering is different, or give different results, than the correlation network, as both are fed with the same information, except for the derivation that can be seen as a pre-processing step as in many other analytical pipelines. Sentences as in page 13 lines 279: Multiple clusters were recovered... are not precise at describing the outcome. Such sentences are found a lot in the manuscript.

I therefore suggest that the authors carefully worked on the structure of their document to better herald their interesting approach. As it is I would not recommend publication.

6. PLOS authors have the option to publish the peer review history of their article (what does this mean?). If published, this will include your full peer review and any attached files.

Reviewer #1: No

Reviewer #2: No

---

## [Author Response · Author response to Decision Letter 0]

5 Apr 2022

We have responded to the reviewer and editor comments, and the response is uploaded with the manuscript files. We have highlighted all the changes in response to the review in a manuscript with tracked changes. Thank you.

---

## [Decision Letter · Decision Letter 1]

29 Apr 2022

Uncovering in vivo biochemical patterns from time-series metabolic dynamics

PONE-D-21-25919R1

Dear Dr. Arnold,

We’re pleased to inform you that your manuscript has been judged scientifically suitable for publication and will be formally accepted for publication once it meets all outstanding technical requirements.

Kind regards,

Oscar Millet

Academic Editor

PLOS ONE

Additional Editor Comments (optional):

Reviewers' comments:

Reviewer's Responses to Questions

**Comments to the Author**

1. If the authors have adequately addressed your comments raised in a previous round of review and you feel that this manuscript is now acceptable for publication, you may indicate that here to bypass the “Comments to the Author” section, enter your conflict of interest statement in the “Confidential to Editor” section, and submit your "Accept" recommendation.

Reviewer #1: All comments have been addressed

Reviewer #2: All comments have been addressed

2. Is the manuscript technically sound, and do the data support the conclusions?

Reviewer #1: Yes

Reviewer #2: Yes

3. Has the statistical analysis been performed appropriately and rigorously? 

Reviewer #1: I Don't Know

Reviewer #2: Yes

4. Have the authors made all data underlying the findings in their manuscript fully available?

Reviewer #1: Yes

Reviewer #2: Yes

5. Is the manuscript presented in an intelligible fashion and written in standard English?

Reviewer #1: Yes

Reviewer #2: (No Response)

6. Review Comments to the Author

Reviewer #1: The authors have refined the manuscript to provide key details and improve the flow of information for a less specialized reader as requested during the initial review. I found the message much clearer in the revised document which now highlights the innovative aspects of the approach.

Reviewer #2: The revised version is substantially improved and the major points were addressed.

The data and the code are available for the community.

I recommend it for publication.

7. PLOS authors have the option to publish the peer review history of their article (what does this mean?). If published, this will include your full peer review and any attached files.

Reviewer #1: No

Reviewer #2: No

---

## [Editor Report · Acceptance letter]

4 May 2022

PONE-D-21-25919R1 

Uncovering *in vivo* biochemical patterns from time-series metabolic dynamics 

Dear Dr. Arnold:

I'm pleased to inform you that your manuscript has been deemed suitable for publication in PLOS ONE. Congratulations! Your manuscript is now with our production department. 

Kind regards, 

on behalf of

Dr. Oscar Millet 

Academic Editor

PLOS ONE